# Immunotherapies against HER2-Positive Breast Cancer

**DOI:** 10.3390/cancers15041069

**Published:** 2023-02-08

**Authors:** Santiago Duro-Sánchez, Macarena Román Alonso, Joaquín Arribas

**Affiliations:** 1Preclinical & Translational Research Program, Vall d’Hebron Institute of Oncology (VHIO), 08035 Barcelona, Spain; 2Centro de Investigación Biomédica en Red de Cáncer (CIBERONC), 08035 Barcelona, Spain; 3Department of Biochemistry and Molecular Biology, Universitat Autónoma de Barcelona, Campus de la UAB, 08193 Bellaterra, Spain; 4Cancer Research Program, Hospital del Mar Medical Research Institute (IMIM), 08003 Barcelona, Spain; 5Department of Medicine and Life Sciences, Universitat Pompeu Fabra (UPF), 08002 Barcelona, Spain; 6Institució Catalana de Recerca i Estudis Avançats (ICREA), 08010 Barcelona, Spain

**Keywords:** HER2, immunotherapy, resistance, vaccines, CAR-Ts, TCBs

## Abstract

**Simple Summary:**

A wide range of treatments are available for HER2-positive breast cancer, which have greatly improved the prognosis and quality of life of these patients. However, resistance to HER2-targeted or untargeted therapies is common in clinical practice and is associated with metastasis, recurrence, and cancer-related death. To address this clinical need, researchers are exploring immunotherapeutic approaches to completely eradicate tumor cells and prevent tumor relapse and progression. In this review, we focus on how these emerging strategies can overcome current resistance and improve the prognosis of patients who do not respond to standard therapies.

**Abstract:**

Breast cancer is the leading cause of cancer-related deaths among women worldwide. HER2-positive breast cancer, which represents 15–20% of all cases, is characterized by the overexpression of the HER2 receptor. Despite the variety of treatments available for HER2-positive breast cancer, both targeted and untargeted, many patients do not respond to therapy and relapse and eventually metastasize, with a poor prognosis. Immunotherapeutic approaches aim to enhance the antitumor immune response to prevent tumor relapse and metastasis. Several immunotherapies have been approved for solid tumors, but their utility for HER2-positive breast cancer has yet to be confirmed. In this review, we examine the different immunotherapeutic strategies being tested in HER2-positive breast cancer, from long-studied cancer vaccines to immune checkpoint blockade, which targets immune checkpoints in both T cells and tumor cells, as well as the promising adoptive cell therapy in various forms. We discuss how some of these new approaches may contribute to the prevention of tumor progression and be used after standard-of-care therapies for resistant HER2-positive breast tumors, highlighting the benefits and drawbacks of each. We conclude that immunotherapy holds great promise for the treatment of HER2-positive tumors, with the potential to completely eradicate tumor cells and prevent the progression of the disease.

## 1. Introduction

Breast cancer is the most common cancer worldwide for both sexes and ranks first in cancer-related deaths among women (GLOBOCAN, 2021). Despite continuous efforts to develop new therapeutic strategies, 29% of patients with breast tumors relapse and often develop metastatic disease with a poor prognosis. In fact, the 5-year survival rate for patients with metastatic breast cancer is only 22% [1]. Of breast tumors, HER2-positive breast cancer represents approximately 15–20% of all cases and is characterized by the overexpression of the tyrosine kinase receptor HER2, making it one of the most aggressive types.

HER2 is a member of the EGFR receptor family, which drives breast tumor proliferation, survival, and invasiveness. Due to the expression of HER2 on the membranes of epithelial tumor cells, several targeted therapies have been developed over the years, starting with trastuzumab (Herceptin) more than 20 years ago. Trastuzumab is a HER2-directed monoclonal antibody that recognizes an epitope in the extracellular region of the protein [2]. Since its discovery in the 1990s, several agents have been developed and approved for the treatment of HER2-positive breast cancer: monoclonal antibodies, pertuzumab and margetuximab; tyrosine kinase inhibitors, lapatinib, neratinib, and tucatinib, small molecules that bind the intracellular domain of HER2 and block its activity; and antibody-drug conjugates, ado-trastuzumab emtansine (T-DM1) and trastuzumab deruxtecan (T-DXd). A recent review exploring the latest advances in antibody–drug conjugates has been published elsewhere [3]. While the use of these targeted agents has greatly improved the outcomes for patients with HER2-positive breast tumors, many patients still do not respond to therapy or relapse after initial response to treatment, leading to metastasis [4]. In fact, therapy resistance and metastatic dissemination are common in clinical settings, so finding the right therapeutic strategy for these patients remains a clinical need.

Clinical resistance to HER2-targeted therapies is often caused by four mechanisms: diminished binding of agents to the HER2 receptor, hyperactivation of signaling molecules downstream of HER2, signaling through alternative pathways, or failure to trigger an adequate antitumor immune response [5,6]. In the case of antibody–drug conjugates, additional mechanisms of resistance may include defects in internalization, in the endosome–lysosome pathway, drug efflux, or drug cytotoxic action [3]. New treatment approaches aim to overcome resistance by harnessing the patient’s immune system to reinforce self-defense mechanisms against the tumor and prevent cancer progression.

The immune system is essential for defending against the development of tumors, as immune cells constantly search for abnormal cells to eliminate. When the immune system fails to clear cancerous cells, surviving cells grow and form a carcinoma [7]. Immunotherapy, a trend in oncology, aims to restore the immune system’s ability to recognize and eliminate abnormal tumor cells. This rapidly growing field is testing a wide range of agents in virtually all tumors [8]. HER2-positive tumors have traditionally been considered “cold” tumors due to their low mutational burden [9], which partially explains why immunotherapies have not been studied as extensively in these tumors as in “hot” tumors (e.g., melanoma). However, recent evidence suggests that HER2-positive tumors are quite immunogenic [10] and may benefit from immunotherapeutic approaches. As a result, several immunotherapeutic approaches are being developed, both pre-clinically and clinically, to target HER2-positive tumors that have relapsed or not responded to current standard treatments.

In this review, we explore the different immunotherapeutic options under development for HER2-positive breast cancer (Figure 1) and how they can overcome current resistance to HER2-directed therapies. We will start with the well-studied field of tumor vaccines, examining the various formats of HER2-based vaccines. We will also cover immune checkpoint inhibition, which is currently approved for use in melanoma or triple-negative breast cancer, in the context of HER2-positive breast cancer. T cell redirection therapies, such as T cell bispecific antibodies (TCBs) and adoptive cellular immunotherapies, which use immune cells from the patient, are particularly promising approaches. Of these, T cell receptor (TCR)-T cell and chimeric antigen receptor (CAR) therapies will be discussed, which harness the innate ability of lymphocytes to survey and eliminate cancer cells. TCBs and CARs have demonstrated efficacy in liquid tumors and hold great potential for resistant solid tumors. The focus of this review is on how these strategies may help overcome resistance to the established standard-of-care therapies for HER2-positive breast cancer.

## 2. HER2 Vaccines

Cancer vaccines aim to stimulate the immune system against a specific antigen, including self-antigens such as HER2, to generate active immunity against the tumor. Cancer vaccines can be classified into two broad categories based on their function: preventive (e.g., vaccine for human papillomavirus) and therapeutic. We will focus on therapeutic cancer vaccines, two of which are FDA-approved: Sipuleucel-T (Provenge) for prostate cancer and Bacillus Calmette-Guerin (BCG) for bladder cancer. Vaccines are generally safe, cost-effective, require fewer doses than other therapies, and can recruit a wide range of immune cells, generating immunological memory which is often absent in current therapies such as trastuzumab [11]. In cases of resistance to treatment due to HER2 downregulation or HER2 intratumor heterogeneity, vaccines may be beneficial as they can target multiple antigens, preventing the clonal selection of HER2-negative cells that may eventually relapse [11].

### 2.1. Vaccines Based on HER2-Derived Peptides

The most clinically advanced vaccines against HER2 administer HER2-derived peptides intradermally and rely on the activity of professional presenting immune cells, which need to process a peptide through the human leukocyte antigen (HLA) machinery. To date, 24 clinical trials of HER2 vaccines have been reported, starting in 1990, most of which utilize three different parts of the HER2 protein.

The extracellular peptide E75 (HER2 369-377, Nelipepimut-S or NeuVax) is a potent CD8+ T cell epitope with high affinity for HLA-A2 and -A3, which are present in 60% of the Caucasian population [12]. Despite being the most studied cancer vaccine, its efficacy is debatable due to conflicting results. In 2014, a phase II clinical trial demonstrated that, in combination with the immune stimulant GM-CSF, E75 raises HER2-directed immunity and improves disease-free survival (DFS) in disease-free, high-risk patients [13]. Conversely, a previous phase III trial (PRESENT) had shown no improvement in DFS in the intention-to-treat population [14,15]. To shed light on these discrepancies, a recent systematic review demonstrated that E75 vaccines decreased, but modestly, recurrence rate and DFS in most studies performed until 2021, and overall survival was not improved significantly [16]. This implies that as single therapy this vaccine is not potent enough. In an attempt to boost its efficacy, combination treatment with trastuzumab was tested in a phase IIb clinical trial in HER2 1+/2+ patients. Despite being safe, DFS again could not be improved in the intention-to-treat population [17] and the capacity of this vaccine to improve clinical parameters remains unsettled.

Two other regions of the HER2 protein have been tested as vaccines with similar outcomes: the transmembrane peptide GP2 (HER 654-662), which is also presented in HLA-A2 molecules [18], did not offer clinical benefit in phase II trials when measuring rate of reoccurrence in clinically disease-free high-risk patients [19]. You et al. confirmed the safety of this vaccine and the potential to elicit a strong immune response in a systematic review, although its clinical efficacy remains controversial [16]. The intracellular peptide AE37 (HER2 776-790), which bears a modification that boosts antigen presentation by increasing epitope charging [20], can activate both CD8+ and CD4+ cells and is also safe to administer, but lacks evidence of efficacy on its own [19]. As combinatorial approaches are more suitable for these entities, combination with trastuzumab (NCT03014076) or Pembrolizumab (NCT04024800) are ongoing.

The use of B cell-specific epitopes has also been tested in a recent clinical trial that used a B cell epitope of HER2 and showed a modest antitumor effect in heavily pretreated patients with different HER2-positive tumors, including breast [21]. Altogether, peptide vaccines are safe and easy to use, but are likely not potent enough for their use as monotherapy, although given their immune-stimulant features, combination therapies may be encouraged to improve antitumor immunity and overcome resistance to current therapies. Nevertheless, peptide vaccines are limited to a single or few epitopes, HLA restriction, and a short half-life, for which some authors have suggested using delivery vectors, such as liposomes, viruses, or nanoparticles [22].

### 2.2. Vaccines Based on HER2 Large Fragments

Vaccines bearing the whole HER2 protein have both HLA I and II epitopes, overcoming this limitation of peptide vaccines [23]. A 2001 in-human study demonstrated that vaccination with the intracellular part of HER2 containing HLA-II helper peptides is safe and develops specific, long-lasting T cell immunity [24]. Another study from the same group in 2004 further proved that this vaccine promoted humoral and cellular immune responses [25], similar to clinical trials by other groups employing different vaccines: a truncated form of HER2 (1-146, 146HER2) or a HER2-fusion protein containing both the extracellular and part of the intracellular domain [26,27]. Interestingly, Kitano’s study found 146 different HER2 antibodies in 14 patients, although no tumor regression was observed, while Curigliano’s study evidenced an overall clinical benefit rate of 30%, although the median time to disease progression was modest (2.8 to 3.4 months). Finally, a 2012 study tested the whole protein with slight modifications in the intracellular domain (dHER2) in refractory patients to trastuzumab [28]. All 12 patients in the study generated anti-HER2 antibodies and no cardiotoxicity was reported, with overall survival of 92% at day 300. HER2 large fragments seem to be better at generating immune responses in patients than peptide vaccines, although their antitumor efficacy still needs further exploring.

### 2.3. Autologous Cells

Autologous tumor cell vaccines use the own patient’s tumor cells to induce an immune response. These vaccines bear the possibility of multivalency, may be modified to secrete cytokines of choice, and can induce polyclonal responses. However, they are also more expensive, complex to manufacture, and miss the broad-spectrum utility of vaccines. Moreover, some authors suggest that these vaccines may induce autoimmunity due to the presence of endogenous self-antigens in the patient’s tumor cells [29]. A landmark article from 1997 showed that autologous renal carcinoma cells modified to express GM-CSF could effectively stimulate the migration of immune cells to the vaccination site [30]. That same year, a phase I trial suggested the clinical efficacy of autologous cell vaccines in breast and ovarian cancer [31]. However, subsequent studies were not so clear and nowadays the efficacy of this strategy is debatable and is being tested in patients with HER2-positive breast cancer (NCT00880464, NCT00317603).

Autologous antigen presenting cells are also being tested as vaccines for their ability to activate other components of the immune system. A B cell-based vaccine encoding a truncated form of HER2 (Ad-k35HM) demonstrated both cellular and humoral immune responses and suppressed tumor growth in mice [32]. Another study used dendritic cells transfected with an adenovirus expressing the HER2 gene (AdNeuTK) and IL-12 in immunocompetent mice and demonstrated antitumor immunity in 60% of mice, implicating both CD4+ and CD8+ T cells [33]. In humans, vaccines consisting of dendritic cells loaded with HER2 peptides also demonstrated CD4+ and CD8+ immune responses (primary tumors 28.6%; invasive breast cancer 8.3%) [34,35]. Of interest, a vaccine named Lapuleucel-T, which consists of PBMCs activated with a HER2-based antigen, has been tested in patients, with a significant increase in immune responses accompanied by modest therapeutic efficacy [36]. Due to this active immunity generation, autologous antigen presenting cells present as one of the best vaccination strategies for HER2-positive tumors.

### 2.4. Nucleic Acid-Based Vaccines

Vaccines carrying nucleic acid molecules represent the most practical approach, given that they are cost-effective and easy to produce. They are mostly plasmid DNA vaccines encoding HER2 based on an efficient delivery system. In-depth reviews covering the topic have already been published [37,38]. The first in-human trial using the full-length signaling-deficient HER2 in plasmid DNA showed no acute toxicity when combined with IL-2 and GM-CSF, with HER2-specific antibodies detectable after several years in a subgroup of patients, although cellular responses were controversial [39]. Contrarily, another trial showed no measurable responses to HER2 by these types of vaccines [40]. Despite this controversy, the strategy is being explored in more trials: NCT00393783 using the rat HER2, NCT00436254 employing the intracellular domain of HER2, and NCT03384914 comparing the efficacy of DNA vaccines to HER2-pulsed dendritic cell vaccines, all in HER2-positive breast cancer patients.

Given the revolution of mRNA vaccines that the COVID-19 pandemic has initiated, mRNA vaccines against HER2-positive breast cancer are expected to be explored soon. Actually, an mRNA vaccine against breast cancer has already been tested, but using the antigen MUC-1 [41]. RNA-based vaccines may express several epitopes and RNA only needs to be transported into the cytoplasm to be translated, avoiding the need for transport to the nucleus. mRNA can also mediate higher protein expression levels in vivo and in a shorter time frame compared with DNA [42]. Moreover, using RNA avoids the risk of potential integration into the genome, although gene expression is relatively transient [42].

To sum up, HER2-directed cancer vaccines may be useful to generate HER2-specific immune responses for some patients with long-lasting immune memory cells that can act as surveillance for dormant cancer cells that may eventually lead to relapse or metastasis. As their antitumor efficacy as single agents is questioned, we envision these agents as adjuvants for current or future therapies to enhance the antitumor immune response.

## 3. Immune Checkpoint Blockade

Immune checkpoints, PD-1 and CTLA-4, are surface receptors expressed by immune cells to control their activation and proliferation. Immune checkpoint blockade (ICB) is the strategy to block the binding of immune checkpoints to their cognate antigens, PD-L1 and B7/CD80, respectively, with the aim to reactivate T cells inside the tumor. Regulatory agencies have already approved ICB for several malignancies, including triple-negative breast tumors (atezolizumab and pembrolizumab, Schmid, P.; 2018); however, HER2-positive breast cancer had not been studied extensively until recently due to the variety of available therapies and the initial thought that HER2-positive breast cancers were not immunogenic [43]. It is now known that HER2-positive breast cancer can have high levels of TILs [44], and PD-L1 [45,46] and the presence of TILs at diagnosis is prognostic of outcome: the number of TILs is inversely correlated with the probability of recurrence and positively correlated with overall survival in HER2-positive breast cancer [44,47,48].

Moreover, treatment with trastuzumab can upregulate PD-L1 expression in breast tumor cells [49,50,51]. Therefore, the combination of ICB with HER2-targeted antibody therapies seems a rational approach to reactivate T cells inside the tumor and boost antitumor immunity. Interestingly, the combination of the ADC T-DM1 with anti-CTLA4 and anti-PD-1 antibodies improves the efficacy of ICB in immunocompetent mouse models through synergistic activation of CD8+ T cells [51]. In the same fashion, Stagg and colleagues demonstrated that the combination of trastuzumab and anti-PD-1 enhances its antitumor effect in preclinical models [52], while Iwata et al. demonstrated improved efficacy with the new generation of ADC Trastuzumab Deruxtecan [53]. These preclinical data support the combination of ICB with HER2-directed therapies in patients and highlight their potential to overcome current resistance to treatment for infiltrated tumors.

### 3.1. PD-1/PD-L1 Blockade

The axis PD-1/PD-L1 is the most studied ICB in HER2-positive breast cancer. Three agents stand out: the anti-PD-1 antibody pembrolizumab and the anti-PD-L1 antibodies atezolizumab and avelumab.

Pembrolizumab is a highly selective, humanized antibody specific for PD-1, which is already approved for the treatment of triple-negative breast cancer. In HER2-positive breast cancer, the PANACEA trial tested the combination of pembrolizumab + trastuzumab in pretreated HER2-positive metastatic breast cancer patients. Noteworthy, the combination was effective for patients resistant to trastuzumab-based therapies that were positive for PD-L1: 15% had a partial objective response to the combined treatment with no dose-limiting toxicities, while no responses were observed in PD-L1-negative patients [54].

Atezolizumab, a fully human monoclonal antibody, and avelumab, a fully human anti-PD-L1 monoclonal antibody, have more conflicting results in HER2-positive breast cancer. Atezolizumab is also approved for PD-L1-positive triple-negative breast cancer, but a phase II clinical trial (KATE2) testing T-DM1 + atezolizumab vs. T-DM1 alone in trastuzumab-resistant PD-L1-positive HER2-positive advanced breast cancer showed only a modest improvement in progression-free survival and, worryingly, more adverse events, including one treatment-related death [55]. Despite this downfall, a phase III trial (KATE3) is recruiting to further test its efficacy in a larger patient cohort. Avelumab, currently approved for advanced urothelial carcinoma and Merkel cell carcinoma, was analyzed in the JAVELIN trial, which included 26 patients with HER2-overexpressing tumors. Unfortunately, none of the HER2-positive breast cancer patients showed an objective response [56]. For avelumab, a new clinical trial is testing the combination with trastuzumab and vinorelbine (NCT03414658) in progressive HER2-positive breast cancer.

Given these results, new trials are ongoing: combinations with Trastuzumab/Pertuzumab + paclitaxel (NCT03747120), with ADCs such as T-DM1 (NCT03032107) or T-DXd (NCT04042701, NCT03523572), with tyrosine kinase inhibitor tucatinib, and with trastuzumab for metastatic breast cancer (NCT04512261, NCT04789096). As expected, Pembrolizumab is also being tested with experimental agents such as a HER2 vaccine (VRP-HER2) to treat HER2-positive metastatic breast cancer patients (NCT03632941). Preclinical data showed an improved antitumor effect of combining this HER2 vaccine with pembrolizumab, while the phase I trial demonstrated both its safety and the generation of a HER2-specific immune response [57]. Other trials are recruiting to test the efficacy of Atezolizumab as adjuvant for first line therapies in patients with metastatic HER2-positive breast cancer (NCT03125928, NCT03199885, NCT03417544) or in combination with other drugs, with Trastuzumab + Vinorelbine (NCT04759248), with doxorubicin + cyclophosphamide followed by paclitaxel + trastuzumab + pertuzumab (ddAC-PacHP) (NCT03726879), or with a HER2/4-1BB bispecific antibody (NCT03650348). Altogether, these trials will shed light on the promising future of anti-PD-1 therapy for breast cancer treatment in late lines of treatment.

Preclinical studies have shown promising efficacy for anti-PD-1 ICB in HER2-positive breast cancers, although more controversial results have been obtained with anti-PD-L1 agents. We believe that combination strategies with HER2-targeting agents should be supported. A paramount example of this is the combination of four different therapies (a vaccine, an anti-PD-L1 antibody, entinostat, and T-DM1) that is about to be clinically tested [58]. Noteworthy, the proper assessment of validated biomarkers such as PD-L1 via immunohistochemistry or alternative quantitative techniques is necessary to select the patients that will benefit from ICB.

Recently, a combinatorial strategy to administer ICB together with inhibitors of the PI3K/AKT/mTOR pathway has been suggested [59]. This pathway is frequently dysregulated in breast cancer, promoting tumor development, immunosuppression, and resistance to HER2-targeted agents [60]. Activating mutations in the pathway promotes the recruitment of immunosuppressive cells such as Tregs and MDSCs [61], as well as upregulating PD-L1 [62]. The most studied PI3K inhibitor, alpelisib, is already approved for hormone receptor-positive HER2-negative breast cancer. Preclinical evidence supports the combinatorial use of alpelisib together with anti-PD-1 and CDK4/6 inhibitors [63] or paclitaxel [64] in murine models. Studies on the effectivity of PI3K inhibition with ICB in HER2-positive breast cancer are lacking to determine the efficacy of the combination, as is being done in combination with trastuzumab and T-DM1 [65] (NCT04208178). The mTOR inhibitor everolimus is also being tested for HER2-positive metastatic breast cancer in a phase III trial, BOLERO-3, with promising results in patients bearing mutations in the pathway [66]. This inhibitor may also be tested in combination with ICB in HER2-positive breast cancer to enhance the efficacy of the latter in patients with the pathway mutated.

### 3.2. LAG3 Blockade

A combinatorial strategy being studied is additional targeting of LAG3 (lymphocyte activation gene-3) together with PD-1/PD-L1 blockade. LAG3 is another immune checkpoint found on effector T cells and NK cells and is a marker of exhaustion similar to PD-1 [67]. LAG3-positive TILs have been detected in breast cancer patient samples correlating with the HER2-overexpressing subtype [68]. Combination of LAG3 ICB with antiPD-1 therapy has shown synergistic effects, reducing tumor growth and increasing survival in breast cancer-bearing mice also treated with a dendritic cell vaccine [69]. A LAG3 fusion protein in combination with paclitaxel was tested for metastatic breast cancer and showed high response rates [70]. Interestingly, a high proportion of patients with PD1+ TILs also have LAG3+ TILs, encouraging the combinatorial use of both ICBs, especially in resistant patients [68]. Recently, the LAG3 antibody relatlimab has been approved by the FDA for the treatment of resistant melanoma, evidencing the safety of the therapy [71] and offering hope for the treatment of resistant breast cancers.

Altogether, ICB has the potential to become an adjuvant to current and future therapies to reactivate the immune system against the tumor, especially in patients with already infiltrated and PD-L1-positive tumors.

## 4. Bispecific Antibodies

T cell bispecific antibodies (TCBs) are engineered antibodies that redirect T cells to target cancer cells. They consist of two single-chain antibodies of different binding specificities, one that binds to the T cell receptor (TCR) domain and another that binds to a tumor antigen [72,73]. This allows T cells to recognize and kill cancer cells even if they do not express the normal T cell target HLA. TCBs have been shown to be effective in liquid tumors, and one, blinatumomab, has been approved for use in acute lymphoblastic leukemia [74,75]. This approach holds promise for treating relapsed HER2-positive breast cancer patients, as HER2 can be used as the tumor-associated antigen to redirect T cells.

This strategy can be beneficial for patients with relapsed cancer that still have a functional immune lymphocytic compartment after standard therapies. Different immune cells can be targeted by changing the CD3 arm to markers of other immune cells (e.g., CD56 to direct NK cells), but T cells are the most advanced strategy due to their high cytotoxic potential and abundance. For liquid tumors, there is one TCB already approved: blinatumomab, a TCB targeting CD19, an antigen consistently expressed on B-lineage acute lymphoblastic leukemia cells, which has been shown to be more effective than traditional chemotherapy in cases of relapse [76]. This demonstrates the potential for similar strategies in HER2-positive breast cancer patients who have relapsed on current treatments, using HER2 as the tumor-associated antigen to which T cells can be redirected.

### 4.1. HER2 TCBs

The first bispecific antibody targeting HER2 was reported in 2001 by Sen and colleagues, who showed that a HER2-specific TCB could generate cytotoxic T cells that could kill chemotherapy-resistant tumor cells [77]. Many researchers have since studied these agents in preclinical investigations, and several clinical trials are currently investigating their use in patients (see Table 1). Recently, trispecific antibodies against HER2, CD3, and CD28 (a marker of T cells) have been developed and shown to promote antitumor immunity against HER2-positive breast tumors through CD4+-mediated tumor inhibition, highlighting the role of this immune subtype in addition to CD8+ T cells [78]. The rapid development of antibody engineering may lead to the creation of novel agents in the future. A concern with virtually every HER2-targeting therapy is the risk of on-target off-tumor toxicities due to the expression of HER2, albeit at low levels, in healthy tissues. To address this issue, researchers have developed a TCB with both CD3 and HER2 arms masked by unstructured polypeptides (XTEN). These polypeptides sterically avoid unwanted targeting of HER2 outside the tumor and have cleavage sites for proteases that allow the release of the TCB in the tumor microenvironment. The dysregulated protease activity present in tumors compared to healthy tissues may provide the necessary selectivity for safety [79].

Another option to avoid toxicities is targeting an alternative, tumor-specific antigen that is overexpressed in HER2-positive tumors. One such antigen is p95HER2, which is expressed in about 40% of HER2-positive breast tumors [80]. In 2018, our group developed a p95HER2 TCB that showed potent antitumor activity against p95HER2-expressing breast cells and patient-derived tumor xenografts. Compared to a HER2-TCB, the p95HER2 TCB did not affect HER2-expressing non-transformed cells, making it a safer treatment option for a subgroup of HER2-positive tumors, particularly for patients who have experienced toxicity from HER2-targeting therapies [81].

Other interesting bispecific antibodies are those that bind to HER2 and CD47, PD-1 or 41BB (see Table 2) [82]. A bispecific recombinant fusion protein targeting human CD47 and HER2, called IMM2902, is being investigated in clinical trials in patients with HER2-expressing advanced solid tumors. CD47 is an essential component of the innate immune system as it inhibits the phagocytic activity of myeloid cells by binding to SIRPα, which is especially abundant on macrophages [83]. IMM2902 inhibits tumor cell growth by speeding up the endocytosis and degradation of HER2 and enhances the phagocytosis of macrophages against tumor cells by blocking the interaction between CD47 and SIRPα, which acts as a “don’t eat me” signal [84]. On the other hand, preclinical data have shown potent anti-tumor activity with the blockade of the PD-1/PD-L1 and HER2 pathways, as well as a bridging effect between T cells and tumor cells, with the first-in-class anti-HER2/PD-1 bispecific antibody IBI315. The ongoing clinical trial will determine if this combination of targeted therapy with immunotherapy enhances antitumor activity through multiple mechanisms of action.

Lastly, dual targeting of HER2 and 4-1BB, an immunoreceptor that strongly enhances T cell proliferation, survival, and activity, will potentially boost the immune antitumor effects locally, reducing systemic toxicities [85].

### 4.2. Other HER2 Bispecific Antibodies

There are other types of HER2 bispecific antibodies, which, instead of having a second arm for the targeting and recruitment of immune cells, target different HER2 epitopes or other ERBB family receptors, such as HER3, with the aim of completely blocking the signaling pathway. There are many ongoing clinical trials with these HER2 x HER2 (NCT04276493, NCT02892123, NCT05380882, NCT04040699, NCT03842085, NCT05320874, NCT03084926) and HER2 x HER3 (NCT04501770, NCT03321981, NCT02912949, NCT04100694, NCT00911898) bispecific antibodies, although these strategies are out of this review’s scope.

## 5. Adoptive Cell Therapy

Adoptive cell therapy (ACT) is a type of cellular immunotherapy that uses the body’s immune cells, usually lymphocytes and specifically T cells, to target and kill cancer cells. This includes T cell receptor (TCR) therapy, tumor-infiltrating lymphocyte (TIL) therapy and chimeric antigen receptor T (CAR-T) cell therapy. These strategies have been successful in treating hematological malignancies and show promise in overcoming resistance to HER2-targeted therapies in the treatment of HER2-positive breast cancer.

### 5.1. Tumor Infiltrating Lymphocytes

Tumor-infiltrating lymphocyte (TIL) therapy is a type of adoptive cellular therapy that involves harvesting lymphocytes that have infiltrated tumors, culturing them in the lab, and amplifying them before infusing them back into the patient. TIL therapy has several advantages in the treatment of solid tumors, including the ability to target a diverse range of antigens, strong tumor-homing ability, and low off-target toxicity due to its diverse T-cell receptor clonality. Nevertheless, the successful application of TIL therapy is currently limited to some tumor types such as melanoma [86] and advanced cervical cancer [87], although some preliminary efficacy has been shown in non-small cell lung cancer [88], colorectal cancer [88,89], and breast cancer [90]. In breast tumors, it has been tested for all subtypes and concluded as a reasonable option for resistant patients [91].

TIL therapy needs to bypass some limitations to be effective, such as lack of persistence in vivo and immune suppression of the harsh TME to achieve tumor control, so co-administration of IL-2 is frequent to activate, support expansion of, and prolong survival of infused T cells [92]. TILs and immune responses in HER2-positive breast cancer have been reviewed by another publication on this issue [93]. Briefly, TILs from HER2-positive breast cancer can be expanded ex vivo [91] with around 20% of central memory T cells, which were reactive to autologous tumor cells in vitro and in vivo. In the clinic, an autologous, neoantigen-selected, tumor-reactive TIL product is being tested for patients with advanced solid malignancies, including advanced HER2-positive breast cancer patients who have failed in standard therapies (NCT05576077). Another combinatorial strategy will combine first a dendritic cell vaccine together with trastuzumab to induce CD4+ HER2-specific responses and thus expand TILs. Afterwards, these cells will be extracted and expanded ex vivo and later reinjected in patients with HER2-positive metastatic breast cancer (NCT05378464). This is the kind of rational strategy that is most likely to be beneficial for patients, combining the best of each different immunotherapy.

### 5.2. TCR Engineered Cell Therapies

Engineered TCR-T cell immunotherapy involves infusing T cells that have been genetically modified ex vivo to express specific TCRαβ genes that can recognize peptides presented on the tumor cell surface by HLA molecules. TCR-T therapy has shown some success in the treatment of solid tumors, particularly melanoma [94], but it also faces challenges common to other T cell therapies. One challenge is finding specific tumor antigens to target with the TCR. Even when using highly specific antigens, some TCR-T trials have resulted in severe and even fatal toxicities [95,96]. Another challenge is ensuring that the T cells have optimal avidity and fitness to effectively target and survive in the tumor microenvironment. This can be improved by enhancing TCR affinity, providing additional T cell co-stimulation, or using antibodies that block immunosuppressive signals [97].

TCR-T cells have several advantages over CAR-T cells. They can target not only surface receptors but also intracellular proteins presented as peptides on HLA molecules, including unknown neoantigens. They also have increased capacity to penetrate tumors, as CAR-T cells can become retained on the tumor periphery due to saturation of antigen molecules at the outer part of tumors [98,99].

These features make TCR-T cells a promising immunotherapy for solid tumors. While HER2-targeted CAR-T cells are more advanced in clinical development, TCR-T cells have some superiorities, such as increased tumor penetration, that make them a potential alternative for patients in whom HER2-targeted therapies have failed due to dense or inaccessible tumor tissue, or when new antigen targets are needed to boost the immune response.

### 5.3. Chimeric Antigen Receptor (CARs)

Chimeric antigen receptors (CARs) are engineered synthetic receptors that redirect lymphocytes, most commonly T cells, to recognize and eliminate cells expressing a specific target antigen. CAR binding to target antigens is independent of the HLA receptor [100]. CAR-T cell therapy has been successfully used to treat hematological malignancies, showing remarkable efficacy and durable clinical responses. So far, six CAR-T cell therapies have been approved, four targeting CD19 and two targeting BCMA.

The first reference to HER2 CAR-Ts, and CAR-Ts, dates to 1993. Even before naming them CAR-Ts (i.e., T-bodies), Eshhar and colleagues developed the first reported chimeric single chain fragment variable (scFV) receptor to redirect T lymphocytes to HER2-expressing cells [101]. They combined the binding domains of an anti-HER2 antibody with the CD3-zeta (CD3z) signaling domain of a TCR/CD3 complex, constructing a first-generation CAR [102]. A year later, Moritz et al. added a hinge region between the HER2 scFv and the CD3z domain [103]. These first investigations confirmed that T cells could be activated and redirected toward specific antigens independently of the HLA. From these initial designs, CARs have evolved and incorporated several features that improve their antitumor properties, especially to favor effectivity in the treatment of solid tumors. To date, several HER2 CAR-Ts have been tested in clinical settings with various degrees of success [104].

Compared to current treatments, HER2 CAR-Ts offer several advantages that can help to overcome current resistances: a different mechanism of action, the fact that CAR-Ts can penetrate and eradicate tumors inaccessible to antibodies [105], and the possibilities for combinatorial targeting [106,107]. The first clinical trial with a HER2-CAR-T, led by Dr. Seven Rosenberg in 2009, treated metastatic HER2-positive breast cancer (NCT00924287). HER2-CAR-Ts were administered in combination with Aldesleukin (IL-2) after lymphodepleting conditioning. Unfortunately, the first woman treated died due to the CAR-Ts administration, and the study was therefore terminated. The cause of death was on-target off-tumor recognition of HER2 in normal lung cells, leading to multiple organ failure [108].

Targeting the tumor-specific antigen p95HER2 may be a safer alternative to HER2-directed CAR-Ts [81]. Our group recently published a pre-print on the first development of p95HER2 CAR-Ts, which showed promising efficacy in both orthotopic HER2-positive breast tumor models and metastasis [109].

Several clinical trials are currently active testing HER2 CAR-Ts for the treatment of several HER2-positive tumors aside from breast, such as brain malignancies (NCT02442297, NCT01109095, NCT03500991), sarcomas (NCT04995003, NCT00902044), lung cancer (NCT03198052), pleural and peritoneal metastasis (NCT04684459), ependymoma (NCT04903080), and pancreas tumors (NCT01935843). All clinical trials including HER2-CAR-Ts that include the treatment of primary breast cancer tumors or derived metastases are summarized in Table 2.

CAR-T cells face several limitations in their use to treat solid tumors, including poor cell trafficking and infiltration, limited T-cell persistence and exhaustion, the presence of an immunosuppressive tumor microenvironment (TME), and the lack of uniform and universal expression of tumor-associated antigens on tumor cells, as well as antigen heterogeneity. To overcome these challenges, researchers have developed several strategies to improve the effectiveness of CAR-T cells in solid tumors [110,111]. Some of these strategies include modifying the CAR-T cells to enhance their trafficking and persistence, targeting multiple antigens to overcome antigen heterogeneity, and using co-stimulatory molecules or immune checkpoint inhibitors to enhance T-cell activation and persistence in the TME. A summary of the most promising strategies to improve CAR-T cells for solid tumors can be found in the review by Abken et al. [102].

#### 5.3.1. Next Generation and Multi-Antigen CAR-Ts

CAR-T cells designed to target a combination of antigens have several advantages, including increased specificity for malignant cells, mitigation of antigen escape, and the ability to target the tumor and its microenvironment. In fact, combinatorial antigen recognition has been shown to effectively eradicate tumor cells [112]. Tandem CAR-Ts have two different single-chain fragment variable (scFV) domains connected, allowing a single CAR-T cell to recognize multiple tumor antigens. In glioblastoma, targeting both HER2 and IL13R alpha 2 with tandem CAR-Ts enhanced antitumor effects due to the prevention of antigen escape have been demonstrated [107]. This is particularly relevant in the case of HER2 intratumor heterogeneity, which often leads to tumor relapse in patients. The feasibility of targeting HER2 in combination with other antigens has also been shown in animal models [107].

Another promising strategy was developed by Choi and colleagues, who used the tumor-specific antigen EGFR-vIII to direct T cells selectively to the tumor through the CAR and these T cells secrete a bispecific antibody that binds both EGFR and CD3 to recruit T cells to EGFR+ tumor cells within the tumor [113]. In this way, CAR-T cells can both target the delivery of a bispecific EGFR-CD3 antibody and act as targeted killers of EGFR-vIII-expressing cells, overcoming antigen escape. Importantly, no detectable toxicities were reported, which warrants further investigation in clinical studies.

An additional promising approach consists of armoring CAR-Ts to secrete cytokines, antibodies, or other immunomodulatory agents in the tumor microenvironment upon CAR–antigen engagement. For example, dual-specific T cells expressing a HER2 CAR and a TCR specific for the melanocyte protein (gp100), together with an indirect recombinant vaccinia virus expressing gp100, were able to eradicate a variety of large solid tumors, including orthotopic breast tumors in immunocompetent mice expressing human HER2 in the breast and brain [114]. Additionally, CAR -Ts secreting cytokines IL-7 and CCL19 can improve immune cell infiltration and CAR-T survival in the tumor [115]. Another group tested the secretion of IL-12 by CAR-Ts with great success in tumor eradication in mice [116].

Immune checkpoint inhibitors are also attractive molecules to be secreted by CAR-Ts. To prevent T cells’ exhaustion and inactivation, Li et al. developed CAR-Ts secreting immune checkpoint inhibitor anti-PD-1 and showed that it reversed the inhibitory effect of the PD-1/PD-L1 interaction on T cell fitness and prompted complete eradication of the tumors, outperforming the parental CAR-Ts [117]. This strategy was clinically reinforced by a phase I study in malignant pleural diseases with positive results combining PD-1 blockade with mesothelin-targeted CAR-Ts in malignant pleural diseases (median overall survival of 23.9 months) [118].

Combinations may not be limited to two agents only and it makes the improvement options of these therapies endless. Porter and colleagues combined the ability of oncolytic viruses, the immunostimulatory potential of cytokine IL-12, and the immune checkpoint blocker anti-PD-L1 with a BiTE specific for the tumor-specific antigen CD44 variant 6 [119]. All these agents together showed to be more potent than each of the components alone. A similar approach could be tested with HER2 CAR Ts to provide potent and durable antitumor responses. This is likely the near future for non-responding HER2-positive tumors, possibly using p95HER2 as the tumor-specific antigen. Whether these immunotherapies will be given as separate agents or with an all-in-one approach remains to be determined.

Most CAR-Ts are generated by random integration of DNA delivered by viral transduction of T cells. Some authors believe that targeted integration using gene editing techniques can better control the natural expression of the CAR. Putting the CAR under the control of endogenous T cell receptor (TCR) promoter, by targeting the CAR to the TRAC locus, improves the phenotype of the CAR Ts, decreases the tonic signaling, and increases the antitumor effect and persistence in vivo [120].

Another potential advantage of eliminating the endogenous TCR is the generation of universal or allogenic CAR-Ts, eradicating the need to generate patient-specific T cells. This prevents a graft-versus-host response without compromising CAR-dependent effector functions [121]. This thought-provoking strategy to deliver allogenic T cells is already being tested in the clinic for the treatment of hematological malignancies (clinical trials NCT04557436, NCT05377827) and may be useful for heavily-pretreated HER2-positive patients that have dysfunctional T cells.

Another current in the field of next generation CAR T cells is focused on the improvement of the inherent CAR design for optimizing T cell fitness and antitumor responses. T cell activation upon receptor engagement is strong and may lead to T cell exhaustion [122], which is accentuated by the redundancy of CD28 and CD3ζ signaling [123]. Many different strategies have been tested preclinically and have been reviewed elsewhere [110,111]. Just as an example, Feucht et al. calibrated the activation potential of CD28 through a single immunoreceptor tyrosine-based activation motif, balancing effector and memory programs, and improving the therapeutic window of CAR Ts [124,125].

In conclusion, CAR-Ts still face several limitations that can be improved in various ways such as armoring, combinatorial antigens, and improved CAR designs. As CAR-Ts offer different mechanisms of action to the current therapies for HER2-positive breast tumors and they can be easily improved, these agents offer the greatest hope for patients with resistant HER2-positive tumors. Ongoing clinical studies will demonstrate which of these agents will prevail in the future.

#### 5.3.2. Alternative Cell Types

Other non-T-cell leukocytes have been modified to express a CAR, although their development is not yet so advanced [126].

#### 5.3.3. CAR-NKs

Differing from T cells, NK cells are innate immune cells that have a wide variety of activator and inhibitory receptors in their membrane that signal to determine cell killing. NK cells modified to express CARs present some potential advantages over CAR T cells: their allogeneic use, their manufacture from cell lines, their capacity to kill cancer cells through both CAR-dependent and CAR-independent mechanisms, and the, in theory, lower toxicity when administered to patients, especially reduced cytokine release syndrome and neurotoxicity [127].

Several CAR-NKs targeting HER2 have been reported, particularly for the treatment of glioblastoma [128,129,130], demonstrating the feasibility and potential of CAR-NKs targeting HER2 in solid tumors. However, almost all the limitations associated with CAR-T therapy also apply to CAR-NK cells, such as those related to the tumor antigen, CAR design, immunosuppressive tumor microenvironment, and manufacturing. NK cells have a short half-life, which could be an advantage in the case of severe toxicity, but a disadvantage if repeated administrations are needed to achieve durable responses [131]. In addition, universally expressed MHC molecules on nucleated cells can inhibit NK cell function and thereby limit their antitumor potential. Despite these limitations, NK cells may play an important role in future immunotherapeutic strategies for HER2-positive tumors due to their potency and allergenicity [127,132].

#### 5.3.4. CAR-Macrophages

Macrophages are innate immune cells that have the ability to engage in antitumor activity, but they can also stimulate angiogenesis, increase tumor invasion, and mediate immunosuppression [133]. There is significant interest in developing CAR-modified macrophages for cancer immunotherapy due to their ability to infiltrate solid tumors and interact with almost all cellular components [127,134,135]. However, clinical experience with CAR macrophages is currently limited, with only one active interventional trial using autologous anti-HER2 CAR macrophages [136].

#### 5.3.5. Other Synthetic Receptors

Alternatively, T cells can be modified to express SynNotch receptors, which, like CARs, have an extracellular scFv-based antigen binding domain, but possess different transmembrane and intracellular domains. Briefly, SynNotch receptors contain a cytosolic domain that is cleaved upon antigen binding, releasing a protein capable of activating specific target genes’ expression [137,138]. A SynNotch receptor that recognizes a specific priming antigen can be used to locally induce expression of a CAR. This enables efficient and controlled tumor cell killing by targeting multiple imperfect but complementary antigens [139,140]. Moreover, controlling CAR expression with a SynNotch receptor reduces tonic signaling and exhaustion, improving the antitumor potential of T cells [139].

## 6. Factors Influencing Response to Immunotherapies

### 6.1. Tumor Microenvironment

Immuno-oncology is changing from a “cancer cell centric” view to a vision that considers all the other factors in the tumor microenvironment (TME) to combat solid tumors [141]. The complex TME is composed of innate and adaptive immune cells, stromal cells, vasculature, and non-cellular components such as soluble factors, signaling molecules, and the extracellular matrix [142]. The TME plays a critical role in neoplastic transformation, tumor growth, invasion, immune evasion, and therapeutic resistance, and also decreases drug penetration [143], supporting the strategy to target one or more components of the TME to improve cancer therapeutics [144,145]. However, it is also known that TME immune cells are capable of surveilling and killing cancer cells in the early stage of tumor development, so this dual role suggests that re-educating rather than destroying the TME could help to increase the efficacy of immunotherapies [146]. There are many different strategies to target the TME, which include immunotherapies, antiangiogenic drugs, and treatments directed against cancer-associated fibroblasts and the extracellular matrix [147].

Breast cancer possesses a complex immunosuppressive TME, as breast tumors often recruit myeloid-derived suppressor cells (MDSCs) that impair both T-cell activation and infiltration [148]. There is limited characterization of the immune TME in HER2-positive tumors, and a more extensive depiction could improve the selection of the therapies [149]. Response rates to ICI are relatively low compared to other tumor types [150]. Given this particularly immunosuppressive TME, any of the immunotherapeutic strategies mentioned in this review could potentially benefit from combination with TME modulators.

Inhibiting MDSCs is an attractive combinatorial approach for increasing the efficacy of HER2 immunotherapies, as it favors T cell infiltration and functionality of innate immune cells. Combination of anti-PD-1 and anti-CTLA-4 with entinostat, a histone deacetylase inhibitor, decreased suppression by granulocytic MDSCs in the TME, significantly improving tumor-free survival in HER2 transgenic breast cancer mouse models [148]. HER2 CAR-T cells that co-express the TR2.4-1BB receptor, a novel chimeric costimulatory receptor that targets tumor necrosis factor-related apoptosis-inducing ligand receptor 2 (TR2) expressed on MDSCs, resulted in TME remodeling, increased T cell proliferation, and a superior antitumor effect against breast cancer tumors [151]. In the case of HER2 monoclonal and bispecific antibodies, it has also been described that modulating tumor infiltrating myeloid cells with agents such as dexamethasone [152], IL4 neutralizers [153], or the small molecule receptor tyrosine kinase cabozantinib [154] can enhance bispecific antibody-driven T cell infiltration and anti-tumor response by reshaping the TME in murine models.

Regulatory T cells (Tregs) also represent a major mechanism of tumor-induced immune suppression by inhibiting effector T cells and altering the entire tumor immune milieu. However, these cells are not so easy to target, as they are subject to delicate stabilization pathways that are highly dependent on the inflammatory conditions. Proper modulation of these pathways could also reveal new molecular targets for improving immunotherapy [155].

A different approach is to target the non-cellular components of the TME, such as the intratumoral signaling, transport mechanisms, metabolism, and oxygenation of the tumor [156]. Hyaluronic acid generates a protumorigenic environment by creating high intratumoral pressure, resulting in blood vessel compression and development of hypoxia, epithelial–mesenchymal transition, tumor progression and metastasis, multidrug resistance, and escape from immune system surveillance. Exogenous hyaluronidase administration showed significant antitumor activity in hyaluronic acid-overexpressing tumors, making hyaluronic acid a promising TME target [157].

Blocking the adenosine axis is also one of the preferred ways to modulate the TME for enhancing T-cell function, since adenosine is used by tumors to promote and sustain their growth. Several agents counteracting the adenosine axis have been developed, and pre-clinical studies have demonstrated important anti-tumor activity, alone and in combination with other immunotherapies including ICI and ACT [158]. In the clinic, there is an active first-in human clinical trial combining the anti-PD-1 Pembrolizumab with SRF617 (NCT04336098), an antibody targeting CD39, a critical enzyme for the extracellular breakdown of ATP and production of adenosine. Increased levels of ATP result in immune cell activation and dendritic cell maturation, whereas decrease adenosine levels will lead to T cell proliferation and activation [159].

Mechanical ultrasound-based approaches to disrupt the TME have also been tested in combination with an anti-PD-L1 antibody with positive results, showing superior systemic antitumor immune responses and distant tumor growth suppression [160].

In summary, these findings provide a rationale for combination therapy of HER2 immunotherapies with TME modulators in patients with resistant breast tumors. Aiming to cure these patients, we should keep in mind that targeting cancer cells is as important as targeting the microenvironment components that initiate and support tumor progression and metastases. Unfortunately, only immunocompetent models can recapitulate the complexity of the interaction between cancer cells and their surroundings, and the majority of immunotherapies that use or redirect human immune cells are being tested in immunodeficient mice [161]. More efforts should be made to generate appropriate models in which these complex and relevant interactions are not being ignored [162].

### 6.2. Microbiome

An important extra-tumoral factor that may influence the success of immunotherapy for HER2-positive breast cancer is the microbiota, defined as the compendium of microorganisms living inside the human body. In fact, the microbiota has already been linked to differences in chemotherapy and endocrine therapy success, as the microbiota is known to play a relevant role in the metabolism of estrogen [163]. Metagenomic analyses suggest that breast cancer is related to bacterial dysbiosis in both the gut and the breast. Notably, changes in the bacterial composition may contribute to cancer progression, especially because microbiota may influence the local and systemic immune system [164]. Most articles focus on the gut microbiota, the most extensive of bacterial reservoirs in the human body, although intratumor bacteria have also been detected in breast cancer. In the study by Nejman and colleagues, thousands of different tumors were analyzed and breast cancer showed the richest intratumor microbiome, regarding both tumor and immune cells, mainly consisting of intracellular bacteria [165]. These authors also found different bacterial types in responders compared to non-responders to ICB in melanoma, although not in the total bacterial load. In the field of HER2-positive breast cancer, the gut microbiota may influence response to trastuzumab in preclinical models and in patients. Di Modica et al. evidenced distinct bacterial diversity and abundance in patients that responded to trastuzumab and patients who did not. Intriguingly, fecal transplants from these patients to mice bearing HER2-positive tumors recapitulated the responses observed in the clinic. This seems to imply a causal relationship between the gut microbiota and the response to trastuzumab [166]. Moreover, antibiotic administration impaired the efficacy of trastuzumab in mice due to an impaired immune response: reduced activation of dendritic cells decreased recruitment of CD4-positive cells and granzyme B-positive cells [166], which would discourage the use of antibiotics simultaneous to anticancer therapy. Supporting this notion, a preliminary clinical study demonstrated that antibiotic intake reduced the efficacy of neoadjuvant therapy in breast cancer patients, with a significantly higher rate of pathological complete responses (29.09% vs. 10.20%, *p*  =  0.017) [167]. Altogether, these results reinforce the idea that the microbiota is an important actor in response to HER2-targeted cancer therapies. The field of microbiota research is gaining momentum in the field of HER2-positive breast tumors and interventional therapies to favor the microbiota, ranging from changes in diet, pre- and probiotics administration, or the more refined injection of engineered bacteria, may be encouraged [168].

## 7. Conclusions and Future Directions

Immunotherapies are emerging as a promising approach for the treatment of HER2-positive breast cancer. These therapies harness the body’s own immune system to fight cancer and have shown promising results in both preclinical models and clinical trials. Cancer vaccines can generate memory cells that may prevent the relapse of dormant HER2-positive cells. Immune checkpoint inhibitors may benefit patients with tumors that express high levels of PD-L1, while bispecific antibodies and adoptive cell therapy have shown potential as potent and direct tumor cell killers that can also reactivate the immune system. However, it is unlikely that any single cancer immunotherapy will be able to overcome all of the evasion mechanisms of solid tumors, so combining multiple immunotherapies may be necessary to achieve the best possible antitumor immune response. To truly achieve personalized medicine, new biomarkers will be needed to determine the most effective immunotherapeutic strategy for each patient. In conclusion, the field of immuno-oncology for HER2-positive breast cancer is rapidly expanding and multimodal immunotherapeutic combinations are likely to become the standard of care for these challenging tumors.

## Figures and Tables

**Figure 1 cancers-15-01069-f001:**
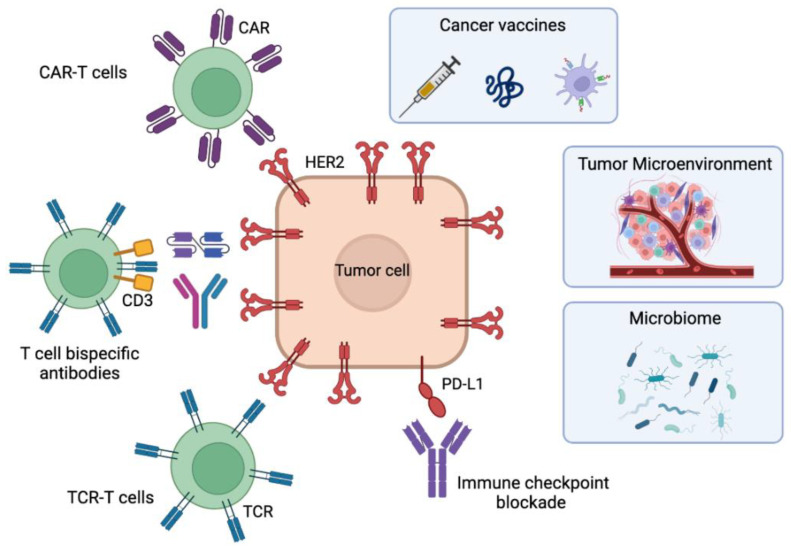
Immunotherapeutic approaches for HER2-positive breast cancer in development to overcome resistances to current therapies. The figure contains cancer vaccines including polypeptide/protein-based or cell-based vaccines, the antibody-based immune checkpoint blockade, the adoptive cell therapies using TCR-T cells, T-cell bispecific antibodies or CAR-T cells, and the factors influencing immunotherapeutic responses: the tumor microenvironment and the microbiome. Figure created with Biorender.com.

**Table 1 cancers-15-01069-t001:** HER2 bispecific antibodies in clinical trials for the treatment of HER2-positive tumors, including breast tumors.

NCT Number	Drug	Target Antigens	Status	Sponsor/Collaborators	Phases	Start
NCT02829372	GBR 1302	HER2 x CD3	Terminated	Ichnos Sciences SA|Glenmark Pharmaceuticals S.A., La Chaux-de-Fonds, Switzerland	Phase 1	May 2016
NCT03983395	ISB 1302	HER2 x CD3	Terminated	Ichnos Sciences SA|Glenmark Pharmaceuticals S.A., La Chaux-de-Fonds, Switzerland	Phase 1|Phase 2	Apr 2020
NCT05076591	IMM2902	HER2 x CD47	Recruiting	ImmuneOnco Biopharmaceuticals Inc., Shanghai, China	Phase 1	Jun 2022
NCT04162327	IBI315	HER2 x PD-1	Recruiting	Innovent Biologics Co., Ltd., Suzhou, China	Phase 1	Nov 2019
NCT03650348	PRS-343	HER2 x 41BB	Active, not recruiting	Pieris Pharmaceuticals, Inc., Boston, MA, USA	Phase 1	Aug 2018
NCT03330561	PRS-343	HER2 x 41BB	Completed	Pieris Pharmaceuticals, Inc., Boston, MA, USA	Phase 1	Sep 2017
NCT05523947	YH32367	HER2 x 41BB	Recruiting	Yuhan Corporation, Seoul, Korea	Phase 1|Phase 2	Aug 2022

**Table 2 cancers-15-01069-t002:** HER2 CAR therapies in clinical trials for the treatment of HER2-positive tumors, including breast tumors and derived metastases.

NCT Number	Treatment	Status	Sponsor/Collaborators	Phases	Start Date
NCT03696030	HER2-CAR T	Recruiting	City of Hope Medical Center|National Cancer Institute (NCI)|California Institute for Regenerative Medicine (CIRM), Duarte, CA, USA	Phase 1	Aug 2018
NCT02713984	HER2-CAR T	Withdrawn	Zhi Yang|Southwest Hospital, Chongqing, China	Phase 1|Phase 2	Mar 2016
NCT02547961	HER2-CAR T	Withdrawn (revision of local regulations)	Fuda Cancer Hospital, Guangzhou, China	Phase 1|Phase 2	Sep 2015
NCT03740256	HER2-CAR T + CAdVEC (oncolytic virus)	Recruiting	Baylor College of Medicine|The Methodist Hospital Research Institute, Houston, TX, USA	Phase 1	Dec 2020
NCT02442297	HER2-CAR T	Recruiting	Baylor College of Medicine|The Methodist Hospital Research Institute, Houston, TX, USA	Phase 1	Feb 2016
NCT04650451	HER2-CAR T with inducible co-activation domain (iMC) and CaspaCIDe^®^ safety switch (BPX-603)	Recruiting	Bellicum Pharmaceuticals, Houston, TX, USA	Phase 1	Dec 2020
NCT04660929	HER2-CAR Macrophages (CT-0508)	Recruiting	Carisma Therapeutics Inc, Philadelphia, PA, USA	Phase 1	Feb 2021
NCT04684459	HER-2/PD-L1 dual-targeting CAR-T	Recruiting	Sichuan University, Chengdu, China	Early Phase 1	Mar 2021
NCT04511871	HER2-CAR T (CCT303-406)	Recruiting	Shanghai PerHum Therapeutics Co., Ltd.|Shanghai Zhongshan Hospital, Shangai, China	Phase 1	Jul 2020
NCT00889954	TGFBeta resistant HER2/EBV-CTLs (EBV-specific cytotoxic T lymphocytes transduced to express the mutant type II TGF-beta dominant-negative receptor and the HER2 CAR)	Completed	Baylor College of Medicine|The Methodist Hospital Research Institute, Houston, TX, USA	Phase 1	May 2009
NCT04430595	4th generation CAR-T cells targeting Her2, GD2, and CD44v6	Recruiting	Shenzhen Geno-Immune Medical Institute|The Seventh Affiliated Hospital of Sun Yat-sen University, Shenzhen, China	Phase 1|Phase 2	Jun 2020
NCT00924287	HER2-CAR T + IV aldesleukin	Terminated (first patient treated on study died as a result of the treatment)	National Cancer Institute (NCI)|National Institutes of Health Clinical Center (CC), Maryland, MD, USA	Phase 1|Phase 2	Nov 2008
NCT03198052	HER2 CAR TS (among other CAR Ts)	Recruiting	Second Affiliated Hospital of Guangzhou Medical University|Hunan Zhaotai Yongren Medical Innovation Co., Ltd.|Guangdong Zhaotai InVivo Biomedicine Co., Ltd., Guangzhou, China	Phase I	Aug 2022

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
