# Peer review of "Immunotherapies against HER2-Positive Breast Cancer"

_cancers, 2023, doi:10.3390/cancers15041069_

Round 1

Reviewer 1 Report

The review is well written and fairly comprehensive, particularly with respect to cellular therapy.  However, there are important gaps in the discussion.

1. Would omit the section on cytokines such as IL2, IL12 and interferon -- these are old subjects and not really being studied currently.

2. Would add a sections on the immune effects of the microbiome, the tumor immune microenvironment, and some of the newer combinations with ICI therapy such as LAG3, PI3K gamma inhibitors, and inhibitors of Tregs and MDSCs.

Reviewer 2 Report

Thank you for letting me review this manuscript. I find the manuscript well written and it is an interesting summary of the field. I have some minor comments:

Stick to HER2-positivity and HER2-negativity instead of HER2+ HER2- for clearness and it makes it easier to read.

The sentence in "Autologous cells" on line 203-206 does not really make sence. Re-phrase.

Line 313 anti-PD-1. And further, stick to the same abbreviation PD-L1 and PD-1 through out the manuscript.

Round 2

Reviewer 1 Report

Thank you for your response.  Minor language changes are suggested:

1. page 8 line 332: change "is also" to "had been"

2. page 8 line335: delete "and worryingly".

3. page 8 line 336: change "downfall" to "concern"

4. page 9 line 377: change "effectivity" to "effectiveness".